# Urine Phenylacetylglutamine Determination in Patients with Hyperphenylalaninemia

**DOI:** 10.3390/jcm10163674

**Published:** 2021-08-19

**Authors:** Fernando Andrade, Ainara Cano, María Unceta Suarez, Arantza Arza, Ana Vinuesa, Leticia Ceberio, Nuria López-Oslé, Gorka de Frutos, Raquel López-Oceja, Elena Aznal, Domingo González-Lamuño, Javier de las Heras

**Affiliations:** 1Metabolomics and Proteomics Platform, Biocruces Bizkaia Health Research Institute, 48903 Bizkaia, Spain; fernando.andradelodeiro@osakidetza.eus (F.A.); raquel.lopezoceja@osakidetza.eus (R.L.-O.); 2Metabolism Group, CIBER-ER, Biocruces Bizkaia Health Research Institute, 48903 Bizkaia, Spain; ainara.canosanjose@osakidetza.eus (A.C.); maria.uncetasuarez@osakidetza.eus (M.U.S.); arantzazu.arzaruesga@osakidetza.eus (A.A.); leticia.ceberiohualde@osakidetza.eus (L.C.); NURIA.LOPEZOSLE@osakidetza.eus (N.L.-O.); gorka.defrutosmunoyerro@osakidetza.eus (G.d.F.); 3Metabolism Section, Biochemistry Laboratory, Cruces University Hospital, 48903 Bizkaia, Spain; 4Division of Pediatric Metabolism, Cruces University Hospital, 48903 Barakaldo, Spain; ana.vinuesajaca@osakidetza.eus; 5Pediatrics Department, Navarra University Hospital, 31008 Pamplona, Spain; elena.aznal.sainz@navarra.es; 6Pediatrics Department, Marqués de Valdecilla University Hospital, 39008 Santander, Spain; Domingo.gonzalez-lamuno@unican.es; 7Department of Pediatrics, University of the Basque Country (UPV/EHU), 48940 Leioa, Spain

**Keywords:** phenylketonuria, phenylalanine, phenylalanine hydroxylase deficiency, phenylacetylglutamine, biomarkers, dried blood spot

## Abstract

Phenylketonuria (PKU), an autosomal-recessive inborn error of phenylalanine (Phe) metabolism is the most prevalent disorder of amino acid metabolism. Currently, clinical follow-up relies on frequent monitoring of Phe levels in blood. We hypothesize that the urine level of phenylacetylglutamine (PAG), a phenyl-group marker, could be used as a non-invasive biomarker. In this cross-sectional study, a validated liquid chromatography coupled to tandem mass spectrometry (LC-MS) method was used for urinary PAG quantification in 35 participants with hyperphenylalaninemia (HPA) and 33 age- and sex-matched healthy controls. We have found that (a) PKU patients present higher urine PAG levels than healthy control subjects, and that (b) there is a significant correlation between urine PAG and circulating Phe levels in patients with HPA. In addition, we show a significant strong correlation between Phe levels from venous blood samples and from capillary finger-prick dried blood spot (DBS) samples collected at the same time in patients with HPA. Further research in order to assess the potential role of urine PAG as a non-invasive biomarker in PKU is warranted.

## 1. Introduction

The deficiency of the enzyme phenylalanine-4-hydroxylase (PAH), which catalyzes the conversion of the essential amino acid phenylalanine (Phe) to tyrosine, results in phenylketonuria (PKU, OMIM 261600) [1]. This autosomal-recessive disease is the most common inborn error of amino acid metabolism. Since PKU was included in newborn screening (NBS) programs, dietary treatment consisting of a low-Phe diet is usually initiated soon after birth to prevent neurological damage and developmental delay. This diet is based on a marked reduction in natural protein, and may require supplementation with Phe-free protein substitutes and specially manufactured low-protein foods [2,3,4]. Early diagnosis by NBS and dietary protein restriction has been successful in preventing neurotoxicity in the majority of patients, but many of them still present suboptimal neurocognitive function [1].

The use of a Phe-restricted diet for PKU treatment goes back 60 years, and varies depending on individual estimated Phe tolerance, which is influenced by other factors such as PAH activity, age, and growth. Therefore, PKU treatment needs constant monitoring and adjustment to achieve appropriate Phe levels and, at the same time, prevent protein catabolism [5]. Poor compliance with therapeutic recommendations is a common phenomenon, especially in chronic pathologies, and the reasons that lead to this behavior are complex and multifactorial, including the patients, the physicians, ineffective communication, and the healthcare systems [6]. General studies of diet adherence in inborn errors of metabolism are limited due to their low prevalence, except for PKU, where it is found that up to 50% of adult patients do not adhere to the prescribed diet [7]. Therefore, several alternative approaches for PKU management have emerged: dietary large neutral amino acid and glycomacropeptide supplementation, pharmacological treatments such as tetrahydrobiopterine, which is effective in a subset of responsive patients, and the recently approved pegylated phenylalanine ammonia lyase [8,9,10].

Since clinical follow-up of PKU patients requires frequent monitoring of Phe levels, as suggested by the European and American PKU Guidelines [11,12], patients have to struggle with sending dried blood spot (DBS) samples with a periodicity which ranges from every month up to every few days, depending on metabolic control and age. Moreover, the severity of the symptoms is not always directly related to circulating Phe levels. Therefore, new strategies have focused on determining novel follow-up biomarkers [13,14,15,16], as additional metabolic markers for PKU could provide a better prediction of the clinical outcome and offer a biochemical explanation for the variability in clinical outcomes observed among treated PKU patients. Over the last decade, untargeted metabolic profiling techniques that evaluate the levels of many metabolites at the same time have emerged as a powerful tool for detecting novel metabolites that can serve as functional biomarkers for inborn errors of metabolism [17]. The molecular structures of Phe-conjugate compounds and intermediates have recently been studied in PKU metabolism, and they may function as biomarkers and contribute to insights into the pathophysiology of the disease [18].

The presence in urine of Phe-compounds, such as phenylacetylglutamine (PAG), could be directly related to increased circulating Phe levels by decarboxylation [19] and could act as a non-invasive surrogate biomarker. PAG is currently used as a biomarker for treatment adherence for patients with urea cycle disorders (UCD) [20]. UCD treatment consists of a protein-restricted diet and, in some cases, essential amino acid supplementation [21]. Additionally, salts and esters of benzoic acid, phenylacetic acid, and phenylbutyric acid have been found to be useful for nitrogen elimination [22]. Benzoic acid is conjugated with glycine to form hippurate, which is excreted in the urine, and phenylbutyric acid is metabolized in the liver to phenylacetyl-CoA, which is then conjugated with glutamine to form PAG and subsequently excreted in the urine. Thus, this compound could be an interesting biomarker for other metabolic diseases such as PKU due to the excretion of its phenyl-groups. PAG excretion was found to be increased in untreated PKU patients forty years ago [23], reflecting secondary endogenous Phe metabolism. Recently, elevated urinary concentrations of Phe-derived catabolites, including phenylpyruvic acid, PAG, and hydroxyphenylacetic acid, were found among non-adherent-to-treatment PKU patients [13].

We hypothesize that urine levels of PAG, a phenyl-group marker, could be used as a non-invasive biomarker for PKU. Therefore, the aim of this cross-sectional study was to determine (a) whether PKU patients present higher urine PAG levels than sex- and age-matched healthy control participants, and (b) to determine whether there is a correlation between urine PAG levels, and Phe levels and dietary protein intake in PKU patients. As there is controversy over the inter-comparability of plasma and DBS Phe results [24,25,26,27,28,29], a secondary aim of this study was to assess the correlation of Phe levels from DBS (capillary finger-prick) and plasma samples obtained at the same time in participants with hyperphenylalaninemia.

## 2. Materials and Methods

### 2.1. Subjects

#### 2.1.1. Patients with Mild Hyperphenylalaninemia (mHPA) and Phenylketonuria (PKU)

In this cross-sectional study, 35 patients with mHPA/PKU from Cruces University Hospital were enrolled. PAH-deficient participants comprised patients with mHPA (Phe concentrations of 120–360 µmol/L and no treatment necessary) (*n* = 6), PKU patients (Phe > 360 µmol/L) treated with BH4 (sapropterin dihydrochloride) (KUVAN^®^, Merck, Madrid, Spain) ± diet (*n* = 12), and PKU patients treated with diet alone (*n* = 17).

The inclusion criteria for the study were: (a) established diagnosis of mHPA or PKU, (b) regular attendance to their scheduled clinical follow-up visits at the Metabolic Unit, and (c) willingness to participate in the study.

Collected data for each patient were age, gender, height, weight, and PKU treatment. In our Metabolic Unit, blood Phe concentrations are regularly measured in HPA/PKU patients for clinical follow-up with the periodicity suggested by the European Guidelines [3,11]. The mean DBS Phe concentration of the previous year was calculated from clinical records.

Recumbent length was measured with a measuring board and weight was determined with a manual baby scale in toddlers up to the age of 24 months. Thereafter, standing height was measured with a wall-mounted stadiometer, and body weight with digital scales. Patients were weighed barefoot after overnight fasting. Height and weight were used to calculate body mass index (BMI).

#### 2.1.2. Control Subjects

Control values for PAG levels in urine were determined in 33 healthy volunteers matched by age and gender.

The study protocol was performed according to the ethical guidelines of the revised 1975 Declaration of Helsinki [30] and approved by the Research Ethics Committee of the Basque Country (CEIm-E), ethics approval code: PI2020236. Written informed consent was obtained from parents or legal guardians of children (below 18 years of age) and adult study participants.

### 2.2. PAG Quantification

#### 2.2.1. Reagents and Solutions

Spot morning urine samples were collected after overnight fasting to measure PAG excretion. The samples were stored at −20 °C until their quantification, normally within a week.

Pure PAG was purchased from Sigma-Aldrich (Madrid, Spain). A deuterated internal standard, PAG-d_5_, was supplied by CDN Isotopes (Quebec, Canada). General reagents and solvents of the purest available quality used in the experimental procedure were supplied by Merck (Madrid) and Teknokroma (Barcelona). In brief, 1 mg/mL stock solutions of all compounds were prepared in ethanol 70% and stored in the dark at 4 °C.

#### 2.2.2. Sample Preparation and Calibration

The samples from the patients and control subjects were analyzed by adding 10 µL of urine and 50 µL of 1 mg/mL for each internal standard in ethanol: water (70:30, *v*/*v*), up to 10 mL. Calibration standards for urine samples were prepared using 10 µL of blank pooled urine, to which 50 µL of 1 mg/mL for internal standard in ethanol: water (70:30, *v*/*v*) were added, and different amounts of 1 mg/mL of PAG in ethanol: water. Pure water was added to the sample in order to obtain a homogeneous total volume of 10 mL for all calibrators. Calibration curves were obtained by the least-square method of relative responses (areas) against concentration of PAG, and concentrations of problem samples were calculated by inverse prediction. PAG levels in urine were also normalized by creatinine, which was quantified by tandem mass spectrometry.

#### 2.2.3. Chromatographic and Instrumental Conditions

The chromatographic system consisted of an Agilent 1100 Series (California, US) equipped with an Agilent Zorbax SB-C18 column (30 × 2.1 mm, 3.5 µm). The chromatography runs with a proportion of methanol of 0.02% *v*/*v* acetic acid 25:75 (*v*/*v*) for 5 min. Flow rate was maintained at 0.3 mL/min during the whole chromatographic run. The tandem mass spectrometer (Agilent 6440 triple quad) used electrospray as an ionization source. The drug and the internal standards were detected in multiple reaction monitoring (MRM) mode, selecting the most intense and selective transitions for each molecule for quantitative purposes in the negative mode. Parameter optimization, reproducibility, and sample stability for this method were discussed by Andrade et al. [20].

### 2.3. Phe Determination

On the day of the visit, capillary blood from a finger-prick was obtained by an experienced nurse at the laboratory of Cruces University Hospital after overnight fasting. Approximately 50 µL of capillary whole blood was used to prepare DBS on Whatman 903 paper cards, allowing it to dry at room temperature. Immediately afterwards, venous blood samples were obtained using lithium heparin tubes. The heparin tube was centrifuged for 10 min at 2000 g and the plasma was aliquoted and stored at −20 °C until the analysis, normally within a week.

DBS samples were analyzed by liquid chromatography-tandem mass spectrometry QSight^®^ 210 MD (PerkinElmer, Waltham, MA, USA) using the PerkinElmer NeoBase™ kit in the Public Health Laboratory of the Basque Government. No correcting factor to convert DBS concentrations to approximate plasma concentrations was used.

The plasma Phe quantification was determined by a Biochrom 30 ion exchange chromatograph (Biochrom, Cambridge, UK) at the Biochemistry Laboratory in Cruces University Hospital using sulfosalicylic acid for deproteinization. After post-column derivatization with ninhydrin, the absorbance was monitored at 570 nm.

### 2.4. Assessment of Diet

Dietary information was collected in a self-administered nutritional dietary intake record over three days (two during the week and one at the weekend) in mHPA/PKU participants. Once completed, protein intake was calculated using the Spanish database *Odimet* [31] by an experienced metabolic dietician.

### 2.5. Statistical Analysis

Differences in quantitative variables between participants with mHPA/PKU and their healthy controls were assessed using Student’s t test, and in categorical variables using χ^2^.

In participants with mHPA/PKU, to assess differences between the three groups (mHPA, PKU patients treated with BH4, and PKU patients treated with diet only), the Kruskal–Wallis test was used to assess differences in quantitative variables, whereas Fisher’s exact test was used to assess differences in qualitative variables.

In all mHPA/PKU participants together (*n* = 35), the Pearson or Spearman correlation—depending on data distribution—was used to assess bivariate relationships between urine PAG/Cr, Phe levels, and protein dietary intake. All statistical assumptions were met. Data are presented as mean ± standard deviation unless otherwise specified. Statistical significance was set at *p* < 0.005. The statistical analysis was performed using SPSS software, version 23, for Windows (IBM, Chicago, IL, USA).

## 3. Results

### 3.1. Urine PAG in mHPA/PKU Patients and Healthy Controls

The study population comprised 35 mHPA/PKU patients and 33 age- and sex-balanced, paired, healthy controls (Table 1, Figure 1).

Participants with mHPA/PKU presented statistically significant higher levels of urine PAG compared to their age- and sex-matched healthy controls (Table 1 and Figure 1).

### 3.2. Characteristics of Patients with mHPA/PKU

Clinical, biochemical, and dietary characteristics of participants with mHPA/PKU are depicted in Table 2. All patients were diagnosed by NBS, except five PKU patients who were born before the NBS program was implemented in Spain.

BH4-non-responsive PKU participants had higher Phe levels (both mean of the last 12 months and on the day of the visit) than BH4-treated PKU and mHPA participants (Table 2).

Regarding diet, there were statistically significant differences in the percentage of patients who were on the Phe-free amino acid mixture between the three groups: (mHPA: 0% vs. BH4-treated PKU: 25% vs. non-BH4-PKU: 100%; *p* ≤ 0.001).

The BH4-non-responsive PKU participants, the group with the highest Phe levels, presented higher urine PAG/Cr levels than BH4-treated PKU and mHPA participants (mHPA: 33.1 ± 21.3 µg/µmol vs. BH4-treated PKU: 34.3 ± 23.3 µg/µmol vs. non-BH4-PKU: 61.5 ± 23.7 µg/µmol; *p* ≤ 0.001).

### 3.3. Correlation between Urine PAG/Cr, Phe Levels, and Protein Dietary Intake in Participants with mHPA/PKU

Urine PAG/Cr significantly correlated with Phe levels (mean of the last 12 months) (R = 0.569, *p* < 0.001), DBS Phe levels on the day of the visit (R = 0.549; *p* < 0.001), and with plasma Phe levels on the day of the visit (R = 0.577; *p* < 0.001) (Table 3 and Figure 2). However, urine PAG/Cr did not significantly correlate with natural dietary protein intake, Phe-free amino acid mixture intake, or with their sum (Table 3).

### 3.4. Correlation between Phe Levels in DBS and Plasma Samples

Phe levels on the day of the visit determined in DBS and in plasma by mass spectrometry and ion exchange chromatography respectively, showed a strong significant positive correlation (R = 0.98; *p* < 0.001) (Figure 3).

## 4. Discussion

Despite the implementation of NBS programs for early detection of PKU in most developed countries, clinical outcomes can be unsatisfactory due to poor adherence to the strict low-protein diet that some patients require, which is difficult to follow at the same time as maintaining normal growth and development. When good metabolic control is not achieved, the essential amino acid Phe can become an intoxicant, despite alternative metabolic pathways that exist to alleviate excessive blood Phe via excretion of aromatic keto acids in urine. Thus, suboptimal clinical outcomes are common in PKU patients and frequent clinical follow-up becomes mandatory.

Clinical follow-up of PKU patients requires frequent monitoring of Phe concentrations, as suggested by the European and American PKU Guidelines [11,12], which is determined in blood in either capillary finger-prick DBS or in venous samples. A new non-invasive reliable biomarker would significantly improve PKU patients’ quality of life and, in this context, urine could offer a non-invasive biological fluid that is better suited for the frequent assessment of Phe metabolism.

Elevated urinary concentrations of Phe and several Phe-derived catabolites, including PAG, were found among PKU patients who did not adhere to their prescribed treatment [13]. In our group, we routinely use urine PAG as a biomarker for treatment adherence in UCD patients [20] and, as urine presence of PAG could be directly related to increased circulating Phe levels by decarboxylation [19], the aim of this study was to evaluate the reliability of PAG as a biomarker in PKU patients using a non-invasive measurement in urine.

In the present study, we used a validated LC-MS method [20] for urinary PAG quantification, and we show that patients with hyperphenylalaninemia (mHPA/PKU) present higher levels of urine PAG than sex- and age-matched healthy controls. Healthy controls excrete a certain amount of PAG, which is comparable to that excreted by mHPA and well-controlled PKU patients, as PAG is physiologically excreted as a result of bacterial metabolism of unabsorbed Phe in the intestinal lumen in the healthy population [23]. Furthermore, we showed that urine PAG/Cr levels significantly correlate with circulating Phe levels, both measured the same day the urine sample was collected, and the mean of Phe levels over the last 12 months. On the contrary, urine PAG levels did not correlate with dietary protein intake. This could be due to the fact that Phe concentration depends not only on protein intake, but also on other important factors, such as PAH activity or growth.

HPA is most commonly caused by pathogenic variants in the *PAH* gene located on chromosome 12, which are inherited in an autosomal-recessive manner. As PKU is genetically very heterogeneous, with >1000 *PAH* variants catalogued in individuals with PKU, leading to more than 2600 known PKU-causing genotypes [1], genotype–phenotype prediction is complex. In this sense, an allelic phenotype value (APV) algorithm has been recently developed, and APV-based phenotype prediction was found to be 99% correct for classic PKU, 46% for mild PKU, and 90% for mHPA [32]. As in our study we have shown a significant correlation between urine PAG and circulating Phe levels, we hypothesize that the *PAH* genotype may also influence urine PAG levels, although this point warrants further research.

In this study, we have classified patients with hyperphenylalaninemia according to the latest recommendations [1,3] into mHPA (Phe concentrations of 120–360 µmol/L and no treatment necessary), PKU patients (Phe > 360 µmol/L) treated with BH4 ± diet, and PKU patients treated with diet alone. In our sample of PKU patients, there was a high prevalence of mild forms, in accordance with our geographical location in the South of Europe [3,33]. In the present study, 12/29 PKU patients received treatment with sapropterin dihydrochloride (BH4). These patients presented lower Phe levels, consumed a higher amount of natural dietary protein, and were less frequently on a Phe-free amino acid mixture than BH4 non-responsive PKU participants (25% vs. 100%). In accordance with their Phe levels, PKU patients not treated with BH4 presented higher urine PAG levels than participants with mHPA and PKU patients treated with BH4.

PAG is formed from phenylacetic acid and Phe, which can competitively inhibit L-amino acid decarboxylase, reducing neurotransmitter levels [34]. Thus, there could be a relationship between elevated PAG excretion and brain Phe and neurotransmitter levels in PKU patients, which warrants further investigation, given the link between neuropsychiatric symptoms, impaired executive function, and neurotransmitter levels in PKU patients [35,36].

Monitoring and adjustments of individual treatments for PKU patients are still based on reliable Phe concentration measurements. Regarding the type of sample for Phe monitoring, DBS sampling is preferred for regular Phe analysis because it allows patients to obtain the sample at home and send it to the laboratory, whereas analysis of a full amino acid profile is considered the gold standard [24]. Despite the concerns about variability of DBS and plasma Phe results shown in previous studies [24,25,26,27,28,29], in this investigation, we have found a strong correlation (R = 0.98, *p* < 0.001) between Phe levels from DBS samples prepared from capillary finger-prick and Phe levels from venous blood without using a correction factor. Although the analytical methods to analyze Phe levels from DBS (tandem mass spectrometry) and plasma (ion-exchange chromatography) were obviously different, pre-analytical factors that can influence Phe level variability were minimized, as all DBS and venous samples were obtained at the same time at Cruces University Hospital by an experienced nurse using the same methodology (spot size, punch location, etc.) and materials (filter paper, puncher, etc.).

A perceived weakness of this study is that, as occurs with rare diseases, the statistical power would improve with a larger cohort of participants. However, we were able to enroll a fairly good number of mHPA/PKU patients and healthy controls, 35 and 33, respectively. Another weakness is that we have not explored neuropsychiatric symptoms and correlated them with urine PAG levels in patients with PKU. In addition, the nutritional records of dietary intake could be subjective and susceptible to under-reporting.

## 5. Conclusions

In this study, we have shown a significant correlation between circulating Phe levels and urine PAG in a cohort of PKU patients. Thus, urine could offer a non-invasive fluid suited for more frequent assessment of Phe metabolism. Nevertheless, further research is warranted in order to assess the potential role of urine PAG as a biomarker in PKU, especially exploring whether high urine PAG levels correlate with neurocognitive outcomes.

## Figures and Tables

**Figure 1 jcm-10-03674-f001:**
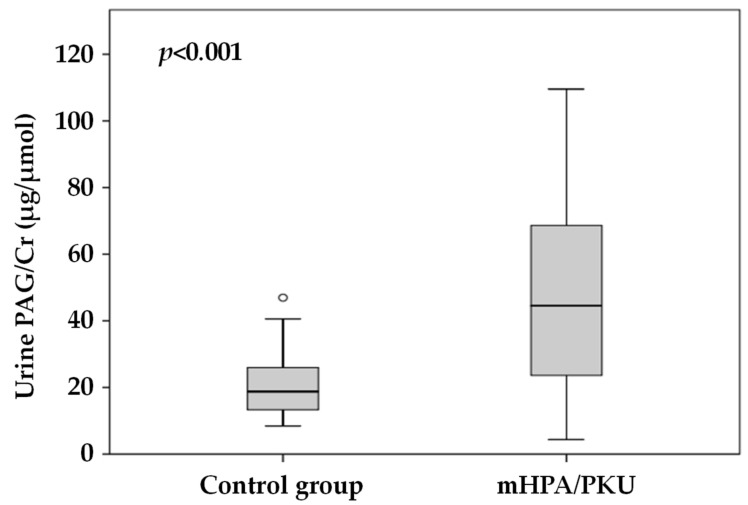
Phenylacetylglutamine concentration in urine from patients with mHPA/PKU (*n* = 35) and their healthy controls (*n* = 33). Mild hyperphenylalaninemia (mHPA), phenylketonuria (PKU), phenylacetylglutamine (PAG), creatinine (Cr).

**Figure 2 jcm-10-03674-f002:**
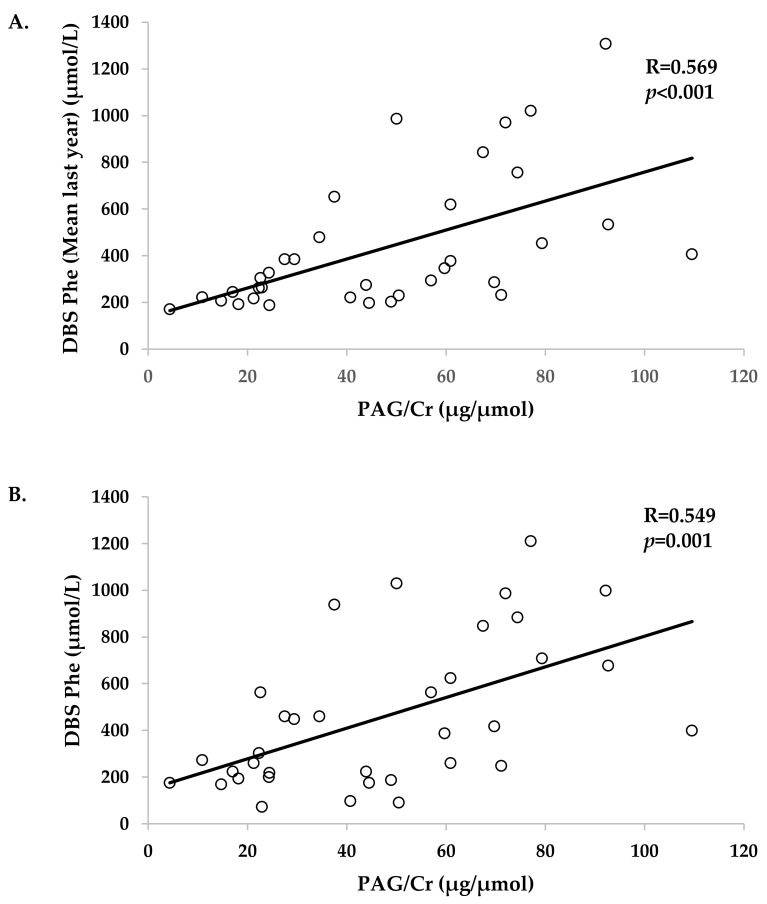
Correlations between urine PAG/Cr and (**A**) mean of the last year DBS Phe levels, (**B**) DBS Phe levels the day of the visit, and (**C**) plasma Phe levels the day of the visit (**C**). Phenylacetylglutamine (PAG), creatinine (Cr), dried blood spot (DBS), phenylalanine (Phe).

**Figure 3 jcm-10-03674-f003:**
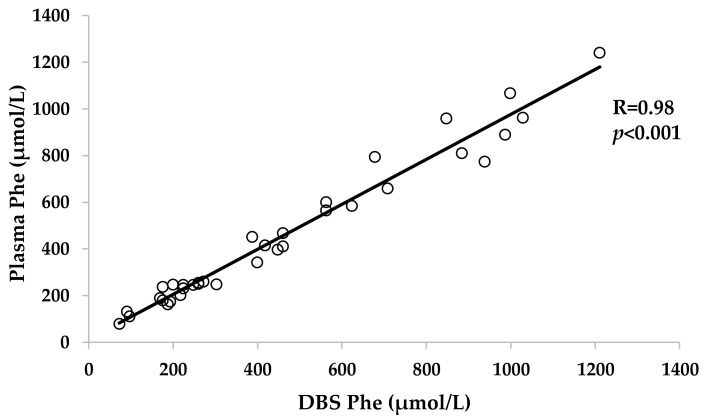
Correlation between the concentration of phenylalanine in plasma and in DBS in mHPA/PKU patients. Mild hyperphenylalaninemia (mHPA), phenylketonuria (PKU), dried blood spot (DBS), phenylalanine (Phe).

**Table 1 jcm-10-03674-t001:** Urine phenylacetylglutamine concentration from mHPA/PKU patients and from their healthy controls.

	Control Group	mHPA/PKU	*p*-Value
*n*	33	35	
Age (years)	18.9 ± 16.6	18.5 ± 15.1	0.683
Gender (male/female)	15/18	15/20	0.829
PAG (µg/mL)	185.2 ± 99.8	403.6 ± 379.1	**<0.001**
PAG/Cr (µg/µmol)	21.3 ± 10.2	47.3 ± 26.5	**<0.001**

Continuous variables are represented as mean ± standard deviation. Mild hyperphenylalaninemia (mHPA), phenylketonuria (PKU), phenylacetylglutamine (PAG), creatinine (Cr). Significant *p*-values are marked in bold.

**Table 2 jcm-10-03674-t002:** Clinical, biochemical, and dietary characteristics of patients with mHPA, BH4-treated PKU, and BH4-non-responsive PKU patients.

	Mild HPA	PKU		*p*-Value
BH4 Treatment	No	Yes	No	
*n*	6	12	17	
Age (years)	10.1 ± 4.3 (5.7–15.3)	14.3 ± 9.3 (1.7–37.4)	24.3 ± 18.5 (0.6–54.8)	0.133
Gender (male/female)	0/6	7/5	8/9	0.104
BMI	19.2 ± 4.9 (15.1–27.1)	22.7 ± 6.0 (15.3–34.4)	21.9 ± 5.6 (14.7–32.1)	0.432
Newborn Screening (yes/no)	6/0	12/0	12/5	**0.019**
DBS Phe (mean last year) (µmol/L)	255.5 ± 70.3 (187.7–385.0)	314.0 ± 163.4 (171.3–756.8)	574.4 ± 337.7 (197.4–1307.7)	**0.012**
DBS Phe (visit day) (µmol/L)	257.3 ± 102.7 (169.5–460.1)	358.7 ± 206.2 (175.6–883.9)	595.8 ± 371.7 (72.7–1210.8)	**0.026**
Plasma Phe (visit day) (µmol/L)	254.5 ± 80.0 (190.0–411.0)	341.6 ± 218.9 (180.0–811.0)	591.1 ± 365.3 (79.0–1241.0)	**0.024**
Urine PAG/Cr (µg/µmol)	33.1 ± 21.3 (14.8–71.2)	34.3 ± 23.3 (4.4–74.4)	61.5 ± 23.7 (22.9–109.6)	**0.006**
Diet				
Natural protein intake (g/kg/day)	2.3 ± 1.2 (1.1–4.2)	1.0 ± 0.6 (0.5–2.6)	0.7 ± 0.2 (0.4–1.0)	**0.001**
Phe-free amino acid mixture (yes/no)	0/6	3/9	17/0	**<0.001**
Phe-free amino acid mixture intake (g/kg/day)	–	0.4 ± 0.3 (0.2–0.7)	0.6 ± 0.2 (0.3–1.0)	0.186
Total protein intake (g/kg/day)	2.3 ± 1.2 (1.1–4.2)	1.1 ± 0.6 (0.8–2.6)	1.3 ± 0.2 (0.8–1.5)	**0.002**

Significant *p*-values are marked in bold. Continuous variables are represented as mean ± standard deviation and range. Mild hyperphenylalaninemia (mHPA), phenylketonuria (PKU), body mass index (BMI), dried blood spot (DBS), phenylalanine (Phe), phenylacetylglutamine (PAG), creatinine (Cr).

**Table 3 jcm-10-03674-t003:** Correlation analysis between urine PAG/Cr and phenylalanine levels and dietary protein intake in participants with mHPA/PKU.

	DBS Phe (Mean Last Year) (µmol/L)	DBS Phe (Visit Day) (µmol/L)	Plasma Phe (Visit Day) (µmol/L)	Natural Protein Intake (g/kg/day)	Phe-free Amino Acid Mixture Intake (g/kg/day)	Total Protein Intake (g/kg/day)
Urine PAG/Cr (µg/µmol)	R = 0.569;	R = 0.549;	R = 0.577;	R = −0.055;	R = 0.287;	R = 0.187;
	***p* < 0.001**	***p* = 0.001**	***p* < 0.001**	*p* = 0.752	*p* = 0.219	*p* = 0.283

Significant *p*-values are marked in bold. Phenylalanine (Phe), phenylacetylglutamine (PAG), creatinine (Cr), mild hyperphenylalaninemia (mHPA), phenylketonuria (PKU), dried blood spot (DBS).

## Data Availability

The data presented in this study are available on request from the corresponding author. The data are not publicly available due to ethical reasons.

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
