# Peer review of "Urine Phenylacetylglutamine Determination in Patients with Hyperphenylalaninemia"

_jcm, 2021, doi:10.3390/jcm10163674_

Round 1

Reviewer 1 Report

The study suggested phenylacetylglutamine (PAG) as a novel non-invasive biomarker for Phenylketonuria patients. In addition, they analyzed correlation between urinary Phe and PAG in patients with mPKU and PKU. The article is well written however there are some concerns that should be clarified:

  1. Phenylpyruvic acid, PAG, hydroxyphenylacetic acid are already known to be upregulated among PKU patients’ indicative of Phe intoxication and hence well studied by other groups (reference 13 DOI: 10.1039/C9AN01642B; which was cited in this manuscript). Further, plasma and urinary Phe are also known to be correlated in PKU patients. Hence the authors claim of PAG being novel PKU biomarker is incorrect. Author cannot claim this research is a novel finding, but again validation of PAG as a PKU biomarker by different research group.
  2. The study analyzed the samples from mPKU, PKU-treated with BH4 and treated with diet alone. PKU is mainly caused due to mutation in the phenylalanine hydroxylase gene. And there are over 1000 mutations each showing a different phenotype with wide epidemiology. The authors should discuss how these mutations affect the levels of PAG.
  3. The headings in Table 2 representing clinical, biochemical and dietary characteristics of patients with mHPA/PKU are a bit confusing. The authors are recommended to mention BH4-treated PKU, BH4-non-responsive PKU and mHPA clearly to avoid confusion.

Author Response

Response to Reviewer 1

The study suggested phenylacetylglutamine (PAG) as a novel non-invasive biomarker for Phenylketonuria patients. In addition, they analyzed correlation between urinary Phe and PAG in patients with mPKU and PKU. The article is well written however there are some concerns that should be clarified:

Thank you very much for your review and insightful comments. We have answered all your questions/comments on a point-by-point reply and believe that our manuscript is much improved with the incorporation of the suggested comments. Our point-by-point response to each comment is detailed below.

Point 1.

Phenylpyruvic acid, PAG, hydroxyphenylacetic acid are already known to be upregulated among PKU patients’ indicative of Phe intoxication and hence well studied by other groups (reference 13 DOI: 10.1039/C9AN01642B; which was cited in this manuscript). Further, plasma and urinary Phe are also known to be correlated in PKU patients. Hence the authors claim of PAG being novel PKU biomarker is incorrect. Author cannot claim this research is a novel finding, but again validation of PAG as a PKU biomarker by different research group.

Response 1.

We agree with the reviewer´s comment, and any claim to novelty has been deleted in the manuscript.

Point 2.

The study analyzed the samples from mPKU, PKU-treated with BH4 and treated with diet alone. PKU is mainly caused due to mutation in the phenylalanine hydroxylase gene. And there are over 1000 mutations each showing a different phenotype with wide epidemiology. The authors should discuss how these mutations affect the levels of PAG.

Response 2.

A new paragraph has been included in the Discussion: “HPA is most commonly caused by pathogenic variants in the PAH gene located on chromosome 12, which are inherited in an autosomal recessive manner. As PKU is genetically very heterogeneous, with >1,000 PAH variants catalogued in individuals with PKU, leading to more than 2,600 known PKU-causing genotypes (1), genotype-phenotype prediction is complex. In this sense, allelic phenotype value (APV) algorithm has been recently developed, and APV based phenotype prediction was found to be 99% correct for classic PKU, 46% for mild PKU and 90% for mHPA (32). As in our study we have shown a significant correlation between urine PAG and circulating Phe levels, we hypothesize that PAH genotype may also influence urine PAG levels, although this point warrants further research.”

Point 3.

The headings in Table 2 representing clinical, biochemical and dietary characteristics of patients with mHPA/PKU are a bit confusing. The authors are recommended to mention BH4-treated PKU, BH4-non-responsive PKU and mHPA clearly to avoid confusion.

Response 3.

The heading in Table 2 has been changed according to the reviewer´s suggestion.

Reviewer 2 Report

This is a very interesting study and introduce a new and non-invasive method to measure phe values. Other researcers within the field of PKU has used biomarkers in urine as an alternative way to monitor phe values (Yano et al. 2013). I do wonder why this present study only include HPA and not moderate or classical patients also? However, this is a start and contribute to overall knowledge about PKU.

Author Response

Response to Reviewer 2

This is a very interesting study and introduce a new and non-invasive method to measure phe values.

Thank you very much for your review and comments. Our point-by-point response to each comment is detailed below.

Point 1.

Other researcers within the field of PKU has used biomarkers in urine as an alternative way to monitor phe values (Yano et al. 2013).

Response 1.

We agree with the reviewer´s comment, and any claim to novelty has been deleted in the manuscript.

Point 2.

I do wonder why this present study only include HPA and not moderate or classical patients also?

Response 2.

In our study we have included patients with mHPA and also patients with PKU. In the study we have classified patients with hyperphenylalaninemia according to the latest recommendations of the European Guidelines into mHPA (Phe concentrations 120-360 µmol/L and no treatment necessary), PKU patients (Phe>360 µmol/L) treated with BH4 ± diet, and PKU patients treated with diet alone.

Point 3.

However, this is a start and contribute to overall knowledge about PKU.

Response 3.

Thank you very much for your kind comment.

Reviewer 3 Report

I think that collection urine samples would be difficult as a matter of storage and shipment methods.

Besides that, it would a good tool to improve clinical management

Author Response

Response to Reviewer 3

I think that collection urine samples would be difficult as a matter of storage and shipment methods.

Besides that, it would a good tool to improve clinical management

Response.

Thank you very much for your review and your kind comments.

Round 2

Reviewer 1 Report

Phenylpyruvic acid, PAG, hydroxyphenylacetic acid are already known to be upregulated among PKU patients’ indicative of Phe intoxication and hence well studied by other groups (reference 13 DOI: 10.1039/C9AN01642B; which was cited in this manuscript). Further, plasma and urinary Phe are also known to be correlated in PKU patients. Hence the authors claim of PAG being novel PKU biomarker is incorrect. Author cannot claim this research is a novel finding, but again validation of PAG as a PKU biomarker by different research group.

Response 1.

We agree with the reviewer´s comment, and any claim to novelty has been deleted in the manuscript.

Minor revision

In conclusion (line 341, 342), "To the best of our knowledge, this is the first time that circulating Phe levels have been correlated with urine PAG in a cohort of PKU patients." should also be modified as this is not the first study correlating Phe and PAG levels in PKU patients. 

After making the above mentioned changes the manuscript can be accepted for publication in JCM.

Author Response

Thank you for your comment. We have changed the conclusion accordingly:

"In this study, we have shown a significant correlation between circulating Phe levels and urine PAG in a cohort of PKU patients..."